# Epithelial-to-Mesenchymal Transition Gene Signature in Circulating Melanoma Cells: Biological and Clinical Relevance

**DOI:** 10.3390/ijms241411792

**Published:** 2023-07-22

**Authors:** Maria Cristina Rapanotti, Elisa Cugini, Elena Campione, Cosimo Di Raimondo, Gaetana Costanza, Piero Rossi, Amedeo Ferlosio, Sergio Bernardini, Augusto Orlandi, Anastasia De Luca, Luca Bianchi

**Affiliations:** 1Department of Anatomic Pathology, University of Rome Tor Vergata, Viale Oxford 81, 00133 Rome, Italy; amedeo.ferlosio@ptvonline.it (A.F.); orlandi@uniroma2.it (A.O.); 2Department of Laboratory Medicine, University of Rome Tor Vergata, Viale Oxford 81, 00133 Rome, Italy; elisa.cgn@gmail.com (E.C.); costanza@med.uniroma2.it (G.C.); bernardini@med.uniroma2.it (S.B.); 3Dermatology Unit, Department of Systems Medicine, University of Rome Tor Vergata, Via Montpellier 1, 00133 Rome, Italy; elena.campione@uniroma2.it (E.C.); cosimodiraimondo@gmail.com (C.D.R.); luca.bianchi@uniroma2.it (L.B.); 4Surgery Division, Department of Surgery Sciences, University of Rome Tor Vergata, Via Montpellier 1, 00133 Rome, Italy; piero.rossi@uniroma2.it; 5Department of Biology, University of Rome Tor Vergata, Via della Ricerca Scientifica 1, 00133 Rome, Italy

**Keywords:** circulating melanoma cells, EMT gene signature, metastasis, responder to therapy

## Abstract

The most promising method for monitoring patients with minimal morbidity is the detection of circulating melanoma cells (CMCs). We have shown that CD45^−^CD146^+^ABCB5^+^ CMCs identify a rare primitive stem/mesenchymal CMCs population associated with disease progression. The epithelial-to-mesenchymal transition (EMT) confers cancer cells a hybrid epithelial/mesenchymal phenotype promoting metastatization. Thus, we investigated the potential clinical value of the EMT gene signature of these primitive CMCs. A reliable quantitative real-time polymerase chain reaction (qRT-PCR) protocol was settled up using tumor cell lines RNA dilutions. Afterwards, immune-magnetically isolated CMCs from advanced melanoma patients, at onset and at the first checkpoint (following immune or targeted therapy), were tested for the level of EMT hallmarks and EMT transcription factor genes. Despite the small cohort of patients, we obtained promising results. Indeed, we observed a deep gene rewiring of the EMT investigated genes: in particular we found that the EMT gene signature of isolated CMCs correlated with patients’ clinical outcomes. In conclusion, We established a reliable qRT-PCR protocol with high sensitivity and specificity to characterize the gene expression of isolated CMCs. To our knowledge, this is the first evidence demonstrating the impact of immune or targeted therapies on EMT hallmark gene expressions in CMCs from advanced melanoma patients.

## 1. Introduction

Approximately 100,000 new cases of melanoma, one of the most aggressive skin cancers, are expected to be diagnosed in 2023, with a death rate of 5% (https://seer.cancer.gov/statfacts/html/melan.html, accessed on 25 June 2023). Despite a favorable prognosis, if surgically excised at early stages, the prognosis becomes severe after metastasization [1]. This is mainly due to the acquisition of chemotherapy resistance and/or to a marginal response to biological therapies, leading to a high mortality rate.

Melanoma progression is articulated in several steps: it implies the transition of benign melanocytic nevi into atypical melanocytic nevi (AMN) that, in turn, can gain a radial growth phase (RGP, horizontal expansion), defined by the horizontal spread of dysplastic melanocytes in the epidermis. Subsequently, atypical nevi could acquire a vertical growth phase (VGP) involving deeper tissues. The transition from RGP to VGP is often accompanied by phenotypic changes conferring melanoma cells’ greater motility. Thus, as soon as melanoma cells cross the epidermal basement membrane, they gain access to blood and lymph vessels, acquiring a great metastatic potential [1,2]. Dissemination of these tumor cells, defined as circulating tumor cells (CTCs), through the bloodstream is considered a pivotal stage of metastatic progression [3,4,5,6,7,8].

In the last years, molecular oncology research has focused on the development of reliable methods to detect, isolate, and then molecularly characterize CTCs, the so-called liquid biopsy. Liquid biopsies fall within the realm of precision and personalized medicine. Melanoma is a cancer type that might greatly benefit from the knowledge associated with liquid biopsy, especially in the administration of therapy, given its heterogeneity. The identification and molecular characterization of circulating melanoma cells (CMCs) in metastatic melanoma patients could allow the monitoring of the disease progression and also give information about its possible response to the available chemotherapy regimens [9,10,11].

In metastatic melanoma (MM), CMCs are detectable in the peripheral blood soon after the surgical resection of the primary tumor—regardless of the thickness—in late-stage disease, or even in clinically disease-free patients [12,13,14,15,16,17]. In 2013, Karakousis and collaborators isolated and counted CMCs, using Melcam and high-molecular-weight melanoma-associated antibodies (HMW-MAAs) as biomarkers, and demonstrated that 26% of MM patients had values ≥ 2 CTCs at baseline. Moreover, the CMCs number inversely correlated with patients’ overall survival (OS), thus functioning as a prognostic factor [18]. This observation was confirmed by two other studies, in which it was found that one or more CMCs at baseline were associated with disease progression of stage IV [19,20] and with shorter relapse-free survival (RFS) of stage III [21,22,23,24]. Additionally, the presence of PDL1^+^ enriched CMCs positively correlated to the patient’s response to pembrolizumab, further strengthening the hypothesis that the molecular characterization of CMCs represents a useful approach for screening patients going through therapies [25].

Our group has established a successful CMCs immuno-enrichment protocol based on the selection of CD146^+^ABCB5^+^ CMCs [26,27,28,29] upon CD45^+^ cell depletion. The cell-adhesion molecule CD146, also known as MCAM (melanoma cell adhesion molecule) or MUC18, is highly expressed at the intercellular junction of endothelial cells, and we have demonstrated that its expression is strongly associated with disease progression [28,29,30,31,32,33]. Of note, the analysis of CD146 molecular expression, at onset or at disease recurrence, could be a useful parameter to follow melanoma remission or progression, even in an apparently disease-free status [26,27,28,29,30,31]. Besides CD146, the ABCB5 trans-membrane transporter is strongly associated with melanoma genesis, stem cell maintenance, metastasis, and chemoresistance [27,28,29,34].

Epithelial-to-mesenchymal transition (EMT) is the main mechanism underlying cancer cell dissemination. It is a multi-step process involving many molecular and cellular changes, including the downregulation of gene coding for epithelial proteins (i.e., *E-cadherin/CDH1*, *occludin/OCLN*, etc.) and the upregulation of gene coding for mesenchymal markers (i.e., *N-cadherin/CDH2*, *fibronectin/hFN1*, *vimentin/VIM*, etc.) [35,36]. Epithelial cancer cells, following the activation of the rewiring of this gene expression, lose their apico-basal polarity and detach from the primary tumor mass acquiring increased motility and invasiveness, which allows them to enter into the bloodstream and, eventually, colonize distant organs [35,37,38,39,40]. EMT is activated upon engagement of different receptors (e.g., TKR, TGFβ family receptors) from several stimuli, including growth factors (i.e., EGF) and cytokines (i.e., TGFβ), which in turn leads to the activation of the EMT transcription factors (EMT TFs) such as TWIST1, ZEB1, SNAI1, and SNAI2 [41]. Interestingly, EMT TFs not only enhance the migratory and invasive properties of cancer cells, but they also guarantee their survival in the bloodstream [35,37,38,39,40,42,43,44,45,46] by supporting their stem phenotype and resistance to apoptotic signals [47,48]. Actually, it has been demonstrated that, rather than in a complete mesenchymal state, cancer cells acquire a hybrid epithelial/mesenchymal phenotype, which is also typical of the majority of CTCs [49] forming distinct cells “subpopulations” characterized by different plasticity, invasiveness, and metastatic potential [50,51]. Of note, it has been found that CD146 is also a strong inducer of EMT in triple-negative breast cancer [52]. However, considering also the pivotal role of CD146 in the metastatization of melanoma [16,26,27,28,30,53], evidence regarding its correlation with EMT hallmarks, in CMCs, is still lacking.

The aim of the present study was to set up and validate a reliable quantitative real-time polymerase chain reaction (qRT-PCR) protocol to describe the epithelial/mesenchymal gene signature of rare CD45^−^CD146^+^ABCB5^+^ CMCs isolated from the blood advanced stage melanoma (III–IV AJCC) patients and to evaluate their potential clinical value. The description of the EMT profile of CMCs may be a useful tool to characterize the aggressiveness of the tumor and to establish appropriate therapies [52,53,54,55,56,57,58,59,60].

To our knowledge, this is the first evidence reporting a preliminary EMT gene signature of CMCs, upon establishment of a reliable protocol for their isolation and molecular characterization, laying the foundations for a future prospective study to establish definitively its clinical relevance.

## 2. Results

### 2.1. Optimization and Validation of qRT-PCR Assay on Control Cell Lines and Healthy Donors

We developed and validated a qRT-PCR assay for the quantitative determination of a panel of EMT genes including the EMT TFs *TWIST1*, *SNAI2*, *ZEB1*, and the EMT hallmarks *CDH1*, *CDH2 HFN1*, *VIM*, and *CD146/MCAM*. First, we analyzed different cell lines: A375 and HeLa cancer cell lines, the fibroblast cell line EDS, the keratinocytes HaCat, the endothelial cell line HUVEC, and the mesenchymal cell lines V126. Then, we tested the specificity of all of the primers by amplifying only the corresponding target amplicon. Thus, we generated complementary DNA (cDNA) using reverse transcriptase from 100 ng of total RNA extracted from our control cell lines. To establish primers calibration curves, we performed 1:5 serial dilutions of the resulting cDNA, starting from 10 ng. Additionally, to accurately examine in vivo isolated CMCs, it was mandatory to determine the correct cycle threshold (Ct) for each gene being analyzed. To this purpose, we performed the analysis on the above-reported control cell lines, and we settled the Ct between 18 and 26. Thus, we set the best Ct, slope, and R^2^ value (>0.96) for each gene target. Genes with a Ct > 42 cycles were considered absent. The results obtained on the control cell lines and the RT-qPCR conditions are reported in Table 1. 

The CMCs enrichment protocol performed on 20 healthy donors did not result in any amplification. Moreover, the highest *EMT TF* expression values obtained from RNAs extracted and amplified from healthy donors were utilized as a “cut-off” to determine positivity.

### 2.2. Detection of In Vivo Isolated CMCs

To investigate the presence of mesenchymal-like CMCs and to evaluate their potential clinical significance, we compared the EMT gene panel expression of CMCs isolated from PB samples collected at onset with those collected at checkpoint/follow-up of melanoma patients undergoing targeted or immune therapy. We performed a preliminary “small” cohort study of seventeen CMCs enrichments from melanoma patients (seven at onset and ten at first follow-up/checkpoint). We enriched the CD45^−^CD146^+^ABCB5^+^ subpopulation of CMCs, as previously reported [27,28,29]. The first relevant result of this study concerns the assessment of the efficiency and the validation of the enrichment assay of the CD45^−^CD146^+^ABCB5^+^ subpopulation of CMCs through RT-qPCR. In particular, we found that the Ct of *ACTB*, which inversely correlates to the amount of isolated and retro-transcripted RNA and thus to the number of CMCs, significantly decreased in patients undergoing disease progression whilst it remained at the onset level in patients in clinical remission (Figure 1a, see below). This observation strongly highlights the relationship between the tumor burden, (i.e., the number of isolated CMCs) and the clinical status of the patients. 

These data were further confirmed by the *ACTB* and EMT hallmark Ct values found in RNAs extracted from healthy donors, which were always below the Ct threshold (Ct > 45). Additionally, compared with healthy donors, the isolated CD45^−^CD146^+^ABCB5^+^ CMCs expressed at least one EMT transcript at onset or at the first checkpoint.

### 2.3. The Expression of EMT Genes of CD45^−^CD146^+^ABCB5^+^ CMCs Subpopulation Differs in Melanoma Patients Undergoing Treatment

To detect possible variations in the EMT profile occurring during therapy, we compared the EMT gene signature of CD45^−^CD146^+^ABCB5^+^ CMCs purified from patients at “onset” to that obtained from patients at “checkpoint”. The results summarized in Table 2 show the following expression patterns (onset/follow-up): *TWIST1* 14.2–30%; (χ^2^ test ** *p* < 0.007), *SNAI2* 0–10% (χ^2^ test ns), *ZEB1* 85–70% (χ^2^ test ** *p* < 0.007), *CD146M/MCAM* 28.5–10% (χ^2^ test *p* = 0.0007), *CDH2* 71.4–50% (χ^2^ test ** *p* < 0.03), *CDH1* 14.2–30% (χ^2^ test ** *p* < 0.007), *CD146M/CAM* 28.5–10% (χ^2^ test *** *p* < 0.05), *CDH2* 71.4–50% (χ^2^ test ** *p* < 0.007), and *VIM* 100–70% (χ^2^ test **** *p* < 0.0001). None of the patients displayed *hFN1*. No specific interaction was documented. The statistical significance obtained was further confirmed by comparing the mean of the expression of each gene between patients at onset and at follow-up (Figure 1b). In particular, we found a significant decrease at follow-up of the EMT genes *VIM*, *CDH2*, *ZEB1*, and *MCAM/CD146* (Figure 1b, see below).

### 2.4. The EMT Gene Signature of CD45^−^CD146^+^ABCB^+^ CMCs Subpopulation Correlates with the Clinical Progression of the Disease

To investigate the possible prognostic significance of the EMT gene signature, we compared the gene expression obtained from patients with a clinical remission status, defined as “responder to therapy”, with those from patients undergoing disease progression, namely, “non-responder”, after 6 months of treatment. Gene expressions from the two groups, reported in Table 3, were as follows (responders/non-responders): *TWIST1* (0–60%, χ^2^ test **** *p* < 0.0001), *SNAI2* (0–20%, χ^2^ test **** *p* < 0.0001), *ZEB1* (60–80%, χ^2^ test ** *p* < 0.003), *CD146* (0–20%, χ^2^ test **** *p* < 0.0001), *CDH1* (20–40%, χ^2^ test ** *p* < 0.003), *CDH2* (20–80%, χ^2^ test **** *p* < 0.0001), and *VIM* (40–100%, χ^2^ test **** *p* < 0.0001). 

The data obtained were further confirmed by comparing the expression of each EMT-related gene between non-responder and responder patients. Despite the small sample size of patients analyzed in this study, we found a significant decrease of *VIM* and of the EMT TF *TWIST1* (Figure 2a) in the responder patients and, surprisingly, an increase in the expression of the EMT TF *ZEB1*.

To extend our analysis, we included the mean of the expression of each gene determined in patients at onset in the statistical analysis. Interestingly, we found that *VIM* expression decreases significantly in responder patients in comparison to both the onset and non-responder patients. Additionally, the expression of the mesenchymal marker *CDH2* also impressively decreased in the responder patients, almost zeroing in some cases, compared to the onset levels. Of note, we observed a drastic decrease in the non-responder patients in the level the EMT TF *ZEB1* with respect to the onset status (Figure 2b). The increase of *CDH1* expression observed in responder patients is promising, which, as expected, showed an opposite trend compared to *CDH2*, as well as of the EMT TF *TWIST1* (Figure 2c), even if not statistically significant.

Interestingly, we observed the absence of molecular expression of *HNF1* in all patients and at all stages of the disease analyzed: onset, responder, and non-responder patients.

In Table 4, the target gene expressions for each patient at onset and at checkpoint control are reported as DCT*1000.

## 3. Discussion

CTCs (circulating tumor cells) play a crucial role in the metastatic cascade. Once detached from primary tumors, they undergo phenotypic changes that confer them a high metastatic potential. Over the past decade, several pieces of evidence have demonstrated that enumerating and molecularly characterizing CTCs can be effectively used as a non-invasive approach to monitor cancer progression and provide information to clinicians [22,61,62,63,64,65]. This information could help determine the most suitable therapy and assess therapy outcomes [66,67,68]. However, the prognostic value of CTCs exhibiting EMT phenotypes remains unknown in many other cancers [69,70,71,72,73].

Various techniques have been reported for isolating and characterizing CTCs based on their physical properties or cell surface antigens [74]. One of the most common approaches is the epithelial cell adhesion molecule (EpCAM)-based enrichment technique. However, recent studies have shown that techniques relying solely on the expression of cell surface antigens may fail to detect certain CTC subpopulations due to their phenotypic heterogeneity [75,76,77]. Phenotypic alterations commonly found in CTCs are a consequence of the activation of the epithelial-to-mesenchymal transition (EMT) program, which is a dynamic multi-step process crucial in the formation of metastases [43,78,79,80,81]. The goal of the EMT process is to enable the dissemination of epithelial cancer cells by conferring them mesenchymal features. It is well established that EMT is mediated by the aberrant activation of epigenetic pathways leading to the downregulation of epithelial markers, such as E-cadherin, and the upregulation of mesenchymal markers, such as N-cadherin, vimentin, and fibronectin. The process then continues following the activation of transcription factors (EMT TFs) such as SNAI1, SNAI2, ZEB1, and TWIST1, which also control cell–cell adhesion, cell migration, and extracellular matrix (ECM) degradation [23,51]. Vimentin, a member of the intermediate filament family of proteins, is ubiquitously expressed in mesenchymal cells. Increased expression of vimentin in cancer cells is associated with enhanced tumor growth and invasiveness [82]. Furthermore, vimentin expression is correlated with the upregulation of N-cadherin, and it has been recently found that in melanoma, protein kinase C-iota (PKC-ι) activates vimentin while inducing EMT [83]. TWIST1 is a helix-loop-helix protein that plays a role in transcriptional regulation during cell differentiation. Increased expression of TWIST1 has been observed in various types of tumor cells, including prostate, gastric, and breast cancer. Moreover, TWIST1 can repress E-cadherin expression while promoting the upregulation of N-cadherin, inducing the so-called “cadherins switch” [84,85]. 

It should be noted that the EMT program does not operate as a binary switch, flipping cells from a completely epithelial to a fully mesenchymal phenotype. Instead, CTCs exist in a series of hybrid states, maintaining both epithelial and mesenchymal or stem cell properties. The acquisition of these mixed phenotypes allows CTCs to survive, evade immune surveillance, and eventually colonize distant organs [49,50,86,87,88]. The study by Yu et al. [46] provided evidence that breast cancer CTCs exhibit dynamic changes in epithelial and mesenchymal composition. These transitions are regulated by various transcription factors, signaling pathways, and changes in the microenvironment [46]. 

All these data support the use of EMT hallmarks and EMT TFs EMT as potential biomarkers for the molecular characterization of CTCs and the subsequent description of the primary tumor aggressiveness or response to therapy. In our study, we specifically focused on three master EMT TFs: ZEB1, SNAI2, and TWIST1. Previous research has demonstrated the significance of these factors in various aspects of melanoma and other cancers. ZEB1 expression promotes immune escape in melanoma [89] and is involved in the acquisition of resistance to MAPK inhibitors [90]. SNAI2 has been directly implicated in melanoma progression by triggering the activation of the GSK-3β/β-catenin pathway [91,92,93]. Moreover, it has been shown to induce stemness features in thyroid cancer [94]. TWIST1 has also been associated with both melanoma stemness- and tumor-initiating properties [95]. Additionally, it has been found to activate the ERK1/2 signaling leading to melanoma invasion and MMP secretion [96]. Thus, in this preliminary phase of our study, we chose to investigate these specific EMT TFs due to their relevance in immunotherapy and targeted therapies. Furthermore, considering the stem/mesenchymal phenotype of the CMCs subpopulation characterized in the present work, we found their involvement in the modulation of stemness and aggressiveness particularly intriguing.

It is widely known that melanoma cells do not express EpCAM. Indeed, malignant transformation of melanocytes, which originate from the neural crest, has been associated with the acquisition of melanoma-specific cell-surface epitopes, such as MCAM/CD146 and MSCP/NG2, (melanoma-associated chondroitin sulphate), as well as stem cell markers, such as ABCB5 (ATP-binging cassette subfamily member B) and CD271 [26,34,97,98]. Our recent studies demonstrated that the acquisition of CD146 and ABCB5 as melanoma-specific targets allows the isolation of highly primitive rare stem/mesenchymal CMCs (CD45^−^CD146^+^ABCB^+^) associated with disease progression. CD146, a cell-adhesion molecule, is known to be highly expressed on the cell surface and is considered an inducer of EMT. In addition to CD146, the transmembrane transporter ABCB5 is strongly associated with melanoma genesis, stem cell maintenance, metastasis, and chemoresistance [99]. Importantly, it has been demonstrated that analyzing the molecular expression of CD146 at the onset of melanoma, or during disease recurrence, could be a useful parameter for monitoring melanoma remission or progression, even in cases of apparent disease-free status [26].

To investigate the presence of mesenchymal-like CMCs and evaluate their potential usefulness as detection markers, we compared the expression levels of EMT TFs in melanoma patient samples collected at the onset of the disease with those collected during checkpoint follow-up, after six months of either targeted or immune therapy. Our first goal was the establishment of a sensitive, specific, and reliable protocol to characterize, by quantitative real-time polymerase chain reaction (qRT-PCR), the EMT-related gene levels starting from low input RNA, such as those obtained from CMCs. To this purpose, we tested the primers used in the present study on serial RNA dilution obtained from both epithelial and mesenchymal cell lines commercially available. Once defined the best experimental conditions, total RNAs were isolated from CD45^−^CD146^+^ABCB^+^ CMCs and analysed through qRT-PCR to identify their EMT-profile. The strength and efficiency of our CMCs isolation protocol were assessed by the mRNA levels of the *ACTB* gene, which served as the internal control in the qRT-PCR assay. We observed an increased amount of *ACTB* gene mRNA in advanced melanoma patients, both at the onset of the disease and in patients experiencing disease progression after six months of therapy. This finding further confirms that the number of purified CMCs, which is correlated with the mRNA levels of the *ACTB* gene, could be used as a valid and non-invasive approach to monitor disease progression and assess patients’ clinical outcomes.

A further confirmation of the strength of our isolation and qRT-PCR protocols for the EMT-multi-marker profile of CMCs arose from the observation of their mesenchymal traits that confirmed the previously reported stem/mesenchymal phenotype of this CMCs subpopulation [27]. In particular, in CMCs isolated from patients during active disease, besides the downregulation of the *CDH1* gene, coding for the epithelial marker E-cadherin, we identified the upregulation of mesenchymal genes coding for vimentin, N-cadherin, the EMT TF TWIST1, and MCAM/CD146, supporting the EMT activation as the cause for metastasis and resistance to chemotherapy. Surprisingly, in none of our samples did we observe the presence of the *hFN1* gene, although it is well-known to be upregulated during EMT and also to be implicated in resistance to radio therapy in cancer. This phenomenon may be related to the concomitant lack of the *SNAI2* gene in this specific subset of CMCs. Indeed, it has been recently reported that circulating tumor cell migration is also promoted through a fibronectin-dependent mechanism that modulates the activity of the transcription factor SNAI2 [100]. 

Despite the small cohort of patients, we extend our analysis by comparing the EMT gene signature of the CD45^−^CD146^+^ABCB^+^-CMCs isolated at onset to the EMT features of CMCs enriched following six months of immune or targeted therapy. As expected, confirming what was already reported in the literature, we found a significant decrease in almost all of the EMT genes during the follow-up [101,102]. 

Regarding melanoma, there is a limited comprehensive understanding of the prognostic value of CMCs. However, a study conducted by Khoja et al. [19] provided insight into this area. The study revealed that 26% of metastatic melanoma patients had baseline CMCs, and patients with two or more CMCs exhibited a significantly shorter median overall survival (OS) compared to those with fewer than two CMCs (2.6 vs. 7.2 months). Furthermore, an additional study supported the prognostic significance of CMCs for OS, demonstrating that having more than two CMCs per 7.5 mL of blood was associated with reduced survival [21]. These studies utilized the Cell-Search Melanoma Kit, a Veridex platform, which identifies CMCs as cells positive for both MCAM/CD146 and MCSP and negative for CD45 and CD34. Therefore, in order to further explore the potential prognostic value of the EMT gene signature in CD45^−^CD146^+^ABCB^+^ CMCs, we deepened our investigation further by stratifying our follow-up samples based on patients’ clinical outcomes. Intriguingly, we found that in patients responding to therapies, there was a statistically significant decrease in the level of the mesenchymal genes *VIM*, *CDH2* (coding for N-cadherin), and of the EMT TF *TWIST1* and a promising upregulation of the epithelial gene *CDH1* (coding for E-cadherin). These shuffling patterns of EMT genes suggest a potential reduction in the aggressiveness of CMCs and possibly a diminished capacity of these cells to colonize distant organs. Conversely, the observation of an increase in the EMT TF *ZEB1* gene in non-responder patients in comparison to the responder ones was surprising. Indeed, a recent study has revealed a novel role for ZEB1, indicating its involvement as a driver of lineage-specific transcriptional programs that regulate cellular state transitions in melanoma. This regulatory function of ZEB1 promotes the emergence of invasive and/or stem-like states within the tumor. Thus, the oscillations in its levels among melanoma patients at onset and during therapies could be indicative of different cell transition states [103].

## 4. Materials and Methods

### 4.1. Patients and Healthy Donors

The current “case series” was overseen and approved by the ethical local institutional review board (code: prot.0013157/2015) of the University of Rome, Tor Vergata. Written informed consent was obtained from the patients before inclusion in the study. Patients were considered eligible if they had a histological and immune-histochemical (S-100, HMB-45, and MART-1) confirmed diagnosis of malignant melanoma and if staged AJCC ≥ pT1b. Patients’ demographic and clinical characteristics are shown in Table 5. The case series was formed by a small cohort of nine melanoma patients with poor prognosis, with treatment-naïve at onset and at the first checkpoint during immune or targeted therapies, as already reported [26,27,28,29].

We collected “baseline blood draw” from seven patients’ treatment-naïve at the onset and 10 blood checkpoint samples after six months of therapy. In particular, for two patients (UPN1-AV and UPN-7 PME), already undergoing checkpoint-inhibitor treatment and in full disease progression at first observation, it was not possible to collect the “baseline point”. However, they were included in the follow-up group. The patient UPN7-ZF checkpoint control blood draw was not reached at the time of this study, so it was included and analyzed only for the “baseline control”. He has currently started targeted therapy.

Five patients out of nine received anti-PD1/PD1-ligand checkpoint immune therapy, while four patients were eligible for targeted therapy anti-BRAF and anti-MEK (UPN4-FM, UPN7-PN). The clinical history of patients is briefly summarized in Table 1. All patients were enrolled at the Dermatology Department of the University of Rome “Tor Vergata” (Italy).

### 4.2. Selection of Reference Genes Panel

The EMT genes included in the panel were selected according to the knowledge already reported in the literature. We included the key players of the EMT, the genes coding for E-cadherin (*CDH1*) and N-cadherin (*CDH2*), which were involved in the “cadherin switch” during EMT [23,85], and the mesenchymal markers *VIM* (*vimentin*) and *HFN1* (*fibronectin*), defined as the core of EMT [23,51,104]. We added the EMT TFs *TWIST1*, *SNAI2*, and *ZEB1*, which orchestrate the gene expression rewiring occurring during EMT [41]. We also included the endothelial antigen *MCAM*/*CD146* as a marker of melanoma progression and metastasis driver [26]. The *ACTB* gene was used as an internal reference gene.

### 4.3. Cell Lines

To set up a reliable protocol that “mimics” as much the detection of our rare CMC sub-population as possible, we developed and analytically validated a qRT-PCR assay for the quantitative determination of the previously cited EMT hallmarks. To this aim, we performed RT-qPCR using the minimum available amount of total RNA, even rather than 100 ng, extracted from two commercial tumor cell lines: A375 (malignant melanoma) and HeLa (cervix cancer) cell lines. The fibroblast cell line EDS, the keratinocytes HaCat, the endothelial cell line HUVEC, and the mesenchymal cell lines V126 (originating from a bone marrow healthy donor) were included as controls. All cell lines were purchased from the American Type Culture Collection (ATCC, Manassas, VA, USA); they were grown in RPMI-1640 (GIBCO-Thermo Fisher Scientific, Waltham, MA, USA) supplemented with 10% heat-inactivated fetal bovine serum (FBS), 2 mM glutamine, 100 U/mL penicillin, and 100 μg/mL streptomycin (GIBCO-Thermo Fisher Scientific, Waltham, MA, USA) in a humidified atmosphere, at 37 °C, 5% CO_2_. Cells were detached by trypsinization, then centrifuged, washed twice with phosphate buffered saline (PBS), and stored at −70 °C, until RNA extraction.

### 4.4. In Vivo CMCs Enrichment

CMCs immune-magnetic enrichment was performed in 17 blood samples obtained from seven melanoma patients. Fifteen ml of peripheral blood (PB) was collected from seven melanoma patients one week after sentinel lymphadenectomy. After six months (first checkpoint control) 15 mL of PB samples from patients undergoing therapy were collected. CD45^−^CD146^+^ABCB5^+^ CMCs were purified through immune-magnetic selection as previously described [27,28,30,31,32]. Samples were considered “CMCs-positive” following detection of MCAM/CD146^+^ or ABCB5^+^ in CD45^−^ nucleated cells. We included human peripheral blood samples from 20 healthy donors from our transfusion centre as negative controls for CD146 and ABCB5 antigens. Isolated CD45^−^CD146^+^ABCB5^+^ CMCs were then analyzed by quantitative RT-PCR (qRT-PCR) for the expression of some EMT hallmarks i.e., *TWIST1*, *SNAI2*, *SLUG*, *ZEB1*, *HFN1*, *CD146/MCAM*, *CDH1*, *CDH2*, and *VIM*. 

### 4.5. Optimization and Validation of the qRT-PCR Assay for the EMT Profile of Enriched CMCs

Total RNA was extracted from control cell lines and purified CMCs subpopulation with TRIzol Reagent (Invitrogen, Waltham, MA, USA) according to the manufacturer’s instructions using a home-made protocol based on the Chomczyńsky and Sacchi method adapted for RNA extraction from an extremely low number of cells, including selective precipitation supported by glycogen [105]. RNA concentration and integrity were measured with ultraviolet spectrophotometry, using the NanoDrop 2000 (Thermo Fisher Scientific, Waltham, MA, USA) according to the manufacturer‘s instructions.

One hundred nanograms of RNA were reverse transcribed using Moloney murine leukemia virus (MMLV) reverse transcriptase (Promega, Madison, WI, USA). Afterwards, RT-qPCR was carried out based on the SYBR Green chemistry (iTaq™ Universal SYBR^®^ Green Supermix, BIO-RAD, Hercules, CA, USA). 

First, we assessed the analytical specificity and sensitivity of the RT-qPCR assay for the detection of the previously reported gene panel through optimization experiments in the above-mentioned cell lines. To this aim, we produced cDNA by reverse transcription (RT) of 100 ng of total RNA and then performed 1:5 serial dilutions of the resulting cDNA, starting from 10 ng, to obtain calibration curves at 7 points. Specifically, we used 10, 2, 0.4, 0.08, 0.0032, and 0 ng of cDNA, in duplicate, to perform RT-qPCR assays for each investigated gene. Once we defined the calibration curves for each hallmark, we settled on the best threshold (Ct), slope, and R2 (>0.96) value for each gene to be used for further experiments. Moreover, at the end of each RT-qPCR run, we performed the melting curve optimization (60–95 °C) to verify the specificity of the reactions [106].

The RT-qPCR assay for the definition of the EMT profile of isolated CMCs was carried out in 20 μL of final volume containing 5 μL of retro-transcripted cDNA, 5 μM of each primer (Table 2), and 50% SYBR green (Kapa SYBR Fast qPCR kit; Kapa Biosystems, Roche, Wilmington, MA, USA) in a StepOnePlus real-time PCR system (Thermo-Fisher). The gene coding for β-actin, ACTB, was used as an internal reference gene. Duplicate reactions were run for each gene. For each sample, an amplification plot and corresponding dissociation curves were examined. Gene expression normalized to β-actin was calculated using the 2^−ΔΔCt^ method. The primer sequences used in RT-qPCR analysis are listed in Table 6.

### 4.6. Statistical Analysis

For the statistical analysis of the EMT mRNAs detected in enriched CMCs, we first compared the data obtained from two different groups: the onset group, which included seven CMCs enriched samples (collected one week after melanoma patients’ sentinel-lymphadenectomy), vs. the first checkpoint-time group, which included ten CMCs enriched samples purified from nine patients after six months of therapy. Gene expression data were represented through box and whiskers boxplots including the single values obtained from each patient. Statistical significance between the onset and follow-up values, for each gene, was evaluated by Student’s *t*-test. Additionally, the frequency of the expression of each gene, expressed as the percentage of patients expressing it at onset and at follow-up, was analyzed via χ^2^ test. Moreover, to get inside the evolution of the EMT process and its potential correlation to patient clinical outcome, the follow-up samples were further stratified into two distinct groups: responder and non-responder to therapy. Significant differences between these groups were evaluated using Student’s *t*-test. The statistical analysis of the EMT gene expression pattern was also performed among the onset, responder, and non-responder groups and evaluated by one-way ANOVA, followed by Tukey’s ad hoc test. The software used for statistical analysis was GraphPad Prism8.0 software (GraphPad Software, San Diego, CA, USA); a *p*-value < 0.05 was considered significant (* *p* < 0.05; ** *p* < 0.01; *** *p* < 0.001; **** *p* < 0.0001).

## 5. Conclusions

In this preliminary translational study, to our knowledge, we are among the first reporting a pilot EMT gene profile of CD45^−^CD146^+^ABCB5^+^ CMCs obtained from advanced melanoma patients at onset and then both in disease progression and clinical remission *status*. We combined in vivo CMCs detection with multi-marker profiling EMT gene/TFs expressions. Although sample size was small, the results obtained clearly showed that CMCs could be efficiently isolated and characterized using our enrichment and qRT-PCR molecular approach. CMCs isolation was based on immunomagnetic enrichment of CD45 depleted, about 99.98% of leukocytes are depleted, by virtue of melanoma-specific CD146/MCAM and ABCB5 antigen expression. Quantitative analysis revealed that all blood samples contained CMCs at numbers that (i) correlated with the severity of the disease or resistance to therapy and (ii) were sufficient for the detection of the EMT gene and EMT TFs by qRT-PCR. 

The described EMT gene signature holds the potential to serve as a foundation for alternative approaches in the isolation and detection CMCs from clinical samples. Utilizing EMT markers for classifying melanoma aids in the identification of the more aggressive CMCs subpopulation and offers valuable insights for determining appropriate clinical strategies.

These findings could represent a preliminary step towards identifying novel markers that can be utilized for the detection or capture of CMCs, particularly those belonging to mesenchymal-like subpopulations. In the Graphical Abstract reported below we show a schematic representation of the subject matter of what we have exposed.

## Figures and Tables

**Figure 1 ijms-24-11792-f001:**
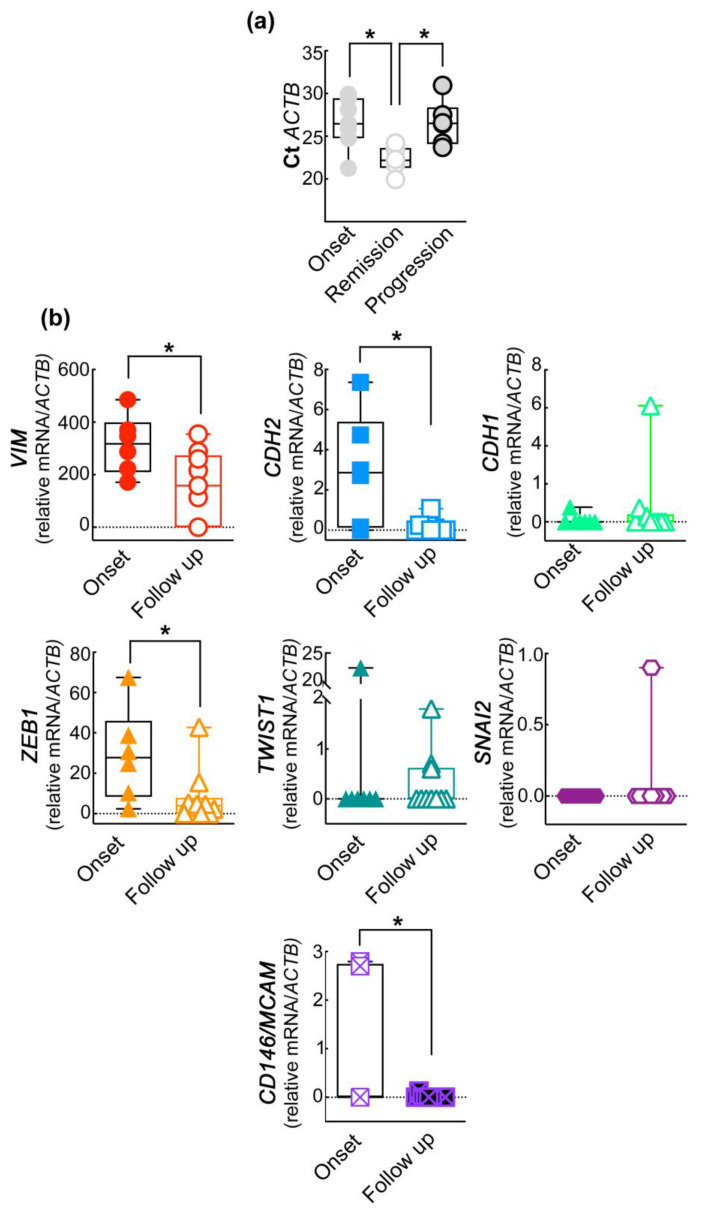
The amplified *ACTB* mRNA and the EMT gene profile of the purified CD45^−^CD146^+^ABCB5^+^ subpopulation of CMCs correlate with patients’ clinical outcomes. (**a**) Analysis of the Ct of the internal reference gene *ACTB* in CD45^−^CD146^+^ABCB5^+^ subpopulation purified from PB of patients at onset and after six months of therapy and clinically classified as in disease remission or progression. One-way ANOVA * *p* < 0.05. (**b**) Expression of the EMT-related genes in CD45^−^CD146^+^ABCB5^+^ CMCs isolated from patients at onset and at follow-up (after six months of therapy). Student’s *t*-test * *p* < 0.05. Data are presented as box and whiskers boxplots including the single values obtained from each patient.

**Figure 2 ijms-24-11792-f002:**
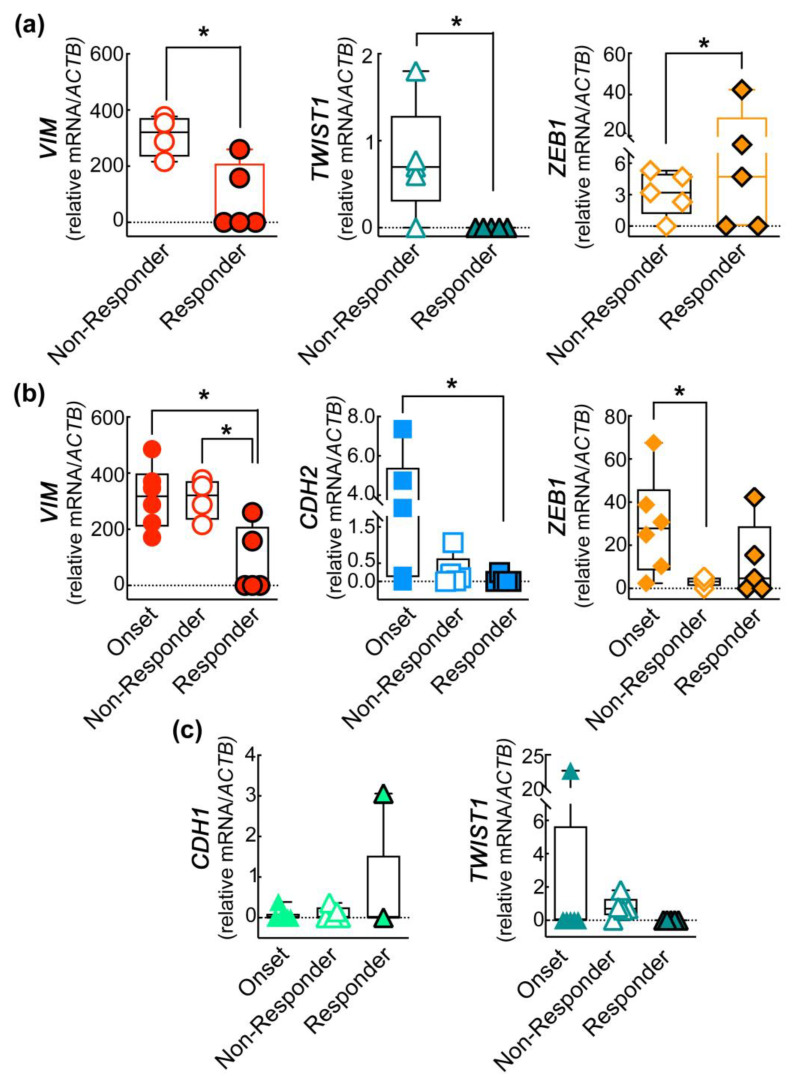
The expression of EMT-related genes in CD45^−^CD146^+^ABCB^+^ CMCs correlates with the clinical progression of the disease. (**a**) Following six months of therapy, patients were divided into responder and non-responder therapy groups. Student’s *t*-test * *p* < 0.05. (**b**) The analysis of the EMT signature in CD45^−^CD146^+^ABCB^+^ CMCs was performed by comparing the gene expression in patients at onset with that found in responder and non-responder patients following six months of therapy. One-way ANOVA * *p* < 0.05. (**c**) The trend of CDH1 and TWIST1 gene expression among the three groups shows promising results. Data are presented as box and whiskers boxplots including the single values obtained from each patient.

**Table 1 ijms-24-11792-t001:** Optimization and validation of RT-qPCR on control cell lines.

Control Cell Line	Gene	Ct Value	Anneal	#Cycles RTqPCR	Thr	Report Standard Curve and Efficiency Curve (90–110%)		Melting Curve (°C)
**V126**	** *SNAI2* **	25.1	60 °C	48	0.2	Slope: −3.19 R^2^: 0.993	105.4%	82
	** *hFN1* **	22	60 °C	48	0.2	Slope: −3.09 R^2^: 0.99	110%	78.5
**A375**	** *n-CADH* **	26.3	60 °C	48	0.3	Slope: −3.37 R^2^: 0.98	97%	78
	** *TWIST1* **	26	60 °C	48	0.3	Slope: −3.4 R^2^: 0.979	94%	89
	** *VIMENTIN* **	18.9	60 °C	48	0.3	Slope: −3.47 R^2^: 0.97	94%	88
	** *ZEB1* **	24.6	60 °C	48	0.3	Slope: −3.33 R^2^: 0.97	99%	81
	** *β-ACT* **	19.07	60 °C	48	0.3	Slope: −3.40 R^2^: 0.96	96%	88
**Hela**	** *MCAM* **	23.88	60 °C	48	0.2	Slope: −3.33 R^2^: 0.998	99%	82
**HaCaT**	** *e-CADH* **	22.0	60 °C	48	0.2	Slope: −3.00 R^2^: 0.99	99%	80.4

Legenda: Annel: annealing temperature; Thr: threshold.

**Table 2 ijms-24-11792-t002:** Molecular expression (percentage) of the EMT gene panel in enriched CMCs subpopulations from melanoma patients at disease onset or after six months follow-up.

Gene	Frequency	Percentage	χ^2^ *p*
*TWIST1*	Onset: 1/7	14%	0.0063
	Follow-up: 3/10	30%	
*SNAI2*	Onset: 1/7	14%	ns
	Follow-up: 2/10	20%	
*ZEB1*	Onset: 6/7	86%	0.0063
	Follow-up: 7/10	70%	
*CD146/MCAM*	Onset: 2/7	29%	0.0007
	Follow-up: 0/10	0%	
*CDH2*	Onset: 5/7	71%	0.0024
	Follow-up: 5/10	50%	
*CDH1*	Onset: 1/7	14%	0.0063
	Follow-up: 3/10	30%	
*VIM*	Onset: 7/7	100%	<0.0001
	Follow-up: 7/10	70%	

**Table 3 ijms-24-11792-t003:** Molecular expression (percentage) of the EMT gene panel in enriched CMC subpopulations obtained from melanoma after six months of follow-up stratified in responders and non-responders to therapy.

Gene	Frequency	Percentage	χ^2^ *p*
*TWIST1*	Responder: 0/5	0%	<0.0001
	Non-Responder: 3/5	60%	
*SNAI2*	Responder: 0/5	0%	<0.0001
	Non-Responder: 1/5	20%	
*ZEB1*	Responder: 3/5	60%	0.002
	Non-Responder: 4/5	80%	
*CD146M/CAM*	Responder: 1/5	20%	<0.0001
	Non-Responder: 0/5	0%	
*CDH2*	Responder: 1/5	20%	<0.0001
	Non-Responder: 4/5	80%	
*CDH1*	Responder: 1/5	20%	0.002
	Non-Responder: 2/5	40%	
*VIM*	Responder: 2/5	20%	<0.0001
	Non-Responder: 5/5	100%	

**Table 4 ijms-24-11792-t004:** Target gene expressions for each patient at onset and at checkpoint control are reported as DCT×1000. For all other patients, the checkpoint control was performed at +6 months. ***** No onset (baseline samples) and two checkpoint controls for both patients were collected for the follow-up analysis.

TARGET GENE (ΔCT×1000) UPN	MCAM//CD146	CDH1	CDH2	TWIST1	SNAI2	ZEB1	VIM	hHFN1	Therapy	Clinical Status
UPN1-AuV *										
1st checkpoint	0	0	0.102	0.664	0	2.309	216.352	0	In treatment	Disease Progression
2nd check point	0	0	0	0	0	0	0	0	Targeted Therapy	Clinical Remission
UPN2-FuM										
Onset	0	0	7.36	0	0	67.485	171.043	0	Naive	
1st checkpoint	0	0	0	0	0	15.441	0	0	Checkpoint Inhibitors	Clinical Remission
UPN3-PaN										
Onset	0	0	0	0	0	38.751	347.245	0	Naïve	
1st checkpoint	0	0	0	0	0	0	113.113	0	Targeted Therapy	Disease Progression
UPN4-CaA										
Onset	2.795	0	4.74	22.581	0	10.298	485.60	0	Naïve	
1st checkpoint	0	0	1.066	0	0.88	4.743	377.357	0	Checkpoint Inhibitors	Disease Progression
UPN5-ZuF										
Onset	0	0	2.721	0	0	30.675	370.746	0	Naïve	
1st checkpoint	0	3.057	0.247	0	0	4.662	260.04	0	Checkpoint Inhibitors	Clinical Remission
UPN6-GiD										
Onset	2.7	0	0.156	0	0	2.377	222.476	0	Naïve	
1st checkpoint	0	0	0	0	0	42.73	158.108	0	Targeted Therapy	Clinical Remission
UPN7-PeME *										
1st checkpoint	0	0	0	0	0	0	0	0	In treatment	Clinical Remission
2nd check point	0.126	0.366	0.207	0.613	0	3.197	354.12	0	Checkpoint Inhbitors	Disease Progression
UPN8-StD										
Onset	0	0.3874	0.04	0	0	1.742	287.527	0	Naive	
1st checkpoint	0	0.146	0.018	1.775	0	5.263	431.553	0	Checkpoint Inhibitors	Disease Progression
UPN9-ZaF										
Onset	0	0	0	0	0	0	79.167	0	Naive	Started to Targeted Therapy

**Table 5 ijms-24-11792-t005:** Patients’ demographic, histological characteristics, and clinical history. * No onset (baseline samples) but two checkpoint controls for both patients were collected for the follow-up analysis. For all other patients, the checkpoint control was performed at + 6 months.

UPN Cod.	Sex	Age at First Observation	Primary Tumor-Site	Histology	Breslow Grade (mm)	AJCC Status at First Observation	Baseline Sample	Therapy	Clinical Status at Follow-Up
UPN1-AuV*	f	80	Unknown	/	/	IV	NO (+2 years)	Targeted Therapy (Anti-BRAF and anti-MEK)	At +12 months: Disease Progression At +18 months: Clinical Remission
UPN2-FuM	m	40	Trunk	SSM	1.25	IIB	YES	Checkpoint Inhibitors (antiPD1-PD1L)	At +6 months: Clinical Remission
UPN3-PaN	m	64	Trunk	NM	4.0	IV	YES	Targeted Therapy (Anti-BRAF and anti-MEK)	At +6 months: Disease Progression
UPN4-CaA	f	82	Extremity	NM	2.4	IV	YES	Checkpoint Inhibitors (anti-PD1-PD1L)	At +6 months: Disease Progression
UPN5-ZuF	m	42	Head	NM	2.2	IB	YES	Checkpoint Inhibitors (anti-PD1-PD1L)	At +6 months: Clinical Remission
UPN6-GiD	m	35	Extremity	SSM	2.2	IIIB	YES	Targeted Therapy (Anti-BRAF and anti-MEK)	At +6 months: Clinical Remission
UPN7-PeME*	f	75	Arm	Mucous M	2.2	IIB	NO (+1 year)	Checkpoint Inhibitors (anti-PD1-PD1L)	At +12 months: Disease Progression At +18 months: Clinical Remission
UPN8-StD	f	41	Trunk	NM	7.0	IIB	YES	Checkpoint Inhibitors (anti-PD1-PD1L)	At +6 months: Disease Progression
UPN9-ZaF	m	42	Extremity	NM	6.0	IVres	YES	Not yet started	Not yet reached

**Table 6 ijms-24-11792-t006:** RT-qPCR primers.

Gene	Primers
*CD146/MCAM*	F: 5′-AGCTCCGCGTCTACAAAGC-3′
	R: 5′-CTACACAGGTAGCGACCTCC-3′
*CDH1*	F: 5′-AAAGGCCCATTTCCTAAAAACCT-3′
	R: 5′-TGCGTTCTCTATCCAGAGGCT-3′
*CDH2*	F: 5′-CTCCTATGA GTGGAA CAG GAA CG-3′
	R: 5′ -TTG GAT CAA TGT CAT AAT CAA GTG CTGTA-3′
*HFN1*	F: 5′-AGCCGAGGTTTTAACTGCGA-3′
	R: 5′-CCC ACT CGGTAAGTGTTCCC-3′
*VIM*	R: 5′-GACGCCATCAACACCGAGTT-3′
	F: 5′-CTTTGTCGTTGGTTAGCTGGT-3′
*SLUG*	R: 5′-CCAAGCTTTCAGACCCCCAT-3′
	F: 5′-GAAAAAGGCTTCTCCCCCGT-3′
*TWIST1*	R: 5′- GCTTGAGGGTCTGAATCTTGCT-3′
	F: 5′-GTCCGCAGTCTTACGAGGAG-3′
*ZEB1*	R: 5′-CAGCTTGATACCTGTGAATGGG-3′
	F: 5′-TATCTGTGGTCGTGTGGGACT-3′
*ACTB*	R: 5′-GAGACCTTCAACACCCCAGCC-3
	F: 5′-AATGTCACGCACGATTTCCC-3′

## Data Availability

The raw data obtained and analyzed during the current study are available from the corresponding authors upon reasonable request.

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
