# Peer review of "Epithelial-to-Mesenchymal Transition Gene Signature in Circulating Melanoma Cells: Biological and Clinical Relevance"

_ijms, 2023, doi:10.3390/ijms241411792_

Round 1

Reviewer 1 Report

In the manuscript entitled "Epithelial to Mesenchymal Transition gene signature in circulating melanoma cells: biological and clinical relevance", the Authors showed qRT-PCR protocol to describe the epithelial-to-mesenchymal gene signature of rare CD45-CD146+ABCB5+ circulating melanoma cells isolated from the blood derived from patients with advanced stage melanoma and  to evaluate their potential clinic value. This manuscript is very interesting, well described, results are novel and within the scope of the journal. However, there are a few issues that should be addressed before publication:

Comment 1: Can the Authors explain why the expression of SNAI1, TWIST2 and ZEB2 are completely omitted? According to the literature data, SNAI1 is responsible for the mesenchymal-related markers overexpression but SNAI2 affects the epithelial-related genes (https://doi.org/10.1186/s12860-021-00356-8; https://doi.org/10.1186/s13046-014-0062-0 and10.1136/thx.2009.121798). In my opinion, this study should be supplemented by the data about the expression of SNAI1, ZEB2 and TWIST2 in all samples.

Comment 2: Font format, size and color should be unified.

Comment 3: Can the Authors explain, why they used only one reference gene in this study and why beta-actin was chosen as a reference gene? Did the Authors check to stability of others reference gene as 18S, B2M or GAPDH? It would be nice if the result of this analyses will be add to this manuscript as supplementary data.

Author Response

To the Attention of Reviewer 1,

Please you wiil find upoladed below our argumentation answer.

Best Regards

Reviewer 2 Report

A manuscript by Rapanotti et al "Epithelial to Mesenchymal Transition gene signature in circulating melanoma cells: biological and clinical relevance" aims at investigating impact of immune- or targeted-therapies on EMT hallmark-gene expressions in CMCs from advanced melanoma patients. Although the general concept of the study is interesting and up-to-date, the study, as also stated by the Authors, too preliminary. A number of patient samples is limited, and the study is rather observational. EMT process is not clearly linked to melanoma, also because of non-epithelial origin of melanoma. Therefore, the concept must be more strongly justified and supported by the results, 

minor revision required

Author Response

To the Attention of Reviewer 2,

Please you wiil find upoladed below our argumentation answer.

Best Regards

Round 2

Reviewer 2 Report

Theo rebuttal letter provided by the Authors is satisfactory.